# Anthocyanins and Type 2 Diabetes: An Update of Human Study and Clinical Trial

**DOI:** 10.3390/nu16111674

**Published:** 2024-05-29

**Authors:** Aleksandra Kozłowska, Aneta Nitsch-Osuch

**Affiliations:** Department of Social Medicine and Public Health, Medical University of Warsaw, 02-106 Warsaw, Poland; aneta.nitsh-osuch@wum.edu.pl

**Keywords:** anthocyanins, diabetes, glucose metabolism

## Abstract

Anthocyanins are phenolic compounds occurring in fruits and vegetables. Evidence from pre-clinical studies indicates their role in glucose level regulation, gut microbiota improvement, and inflammation reduction under diabetic conditions. Therefore, incorporating these research advancements into clinical practice would significantly improve the prevention and management of type 2 diabetes. This narrative review provides a concise overview of 18 findings from recent clinical research published over the last 5 years that investigate the therapeutic effects of dietary anthocyanins on diabetes. Anthocyanin supplementation has been shown to have a regulatory effect on fasting blood glucose levels, glycated hemoglobin, and other diabetes-related indicators. Furthermore, increased anthocyanin dosages had more favorable implications for diabetes treatment. This review provides evidence that an anthocyanin-rich diet can improve diabetes outcomes, especially in at-risk groups. Future research should focus on optimal intervention duration, consider multiple clinical biomarkers, and analyze anthocyanin effects among well-controlled versus poorly controlled groups of patients with diabetes.

## 1. Introduction

Type 2 diabetes mellitus (T2DM) is a significant global health concern that is associated with severe illness and mortality. Over the last thirty years, type 2 diabetes has become significantly more prevalent in countries of all income levels [1,2]. According to the WHO, more than 1.5 million fatalities occur annually as a result of diabetes and diabetes-related complications [3]. In 2021, approximately 353 million adults had diabetes. If secular trends persist, projections indicate that the total number of individuals diagnosed with this disorder will increase to about 578 million by 2030 and 700 million by 2045 [4]. International public health authorities have designated diabetes, along with cardiovascular disease, cancer, and obesity, as diseases requiring immediate concern [5,6].

The progression of T2DM occurs as a result of the interplay between environmental and genetic influences. Several exogenous and endogenous conditions, such as obesity and overweight, a sedentary lifestyle, prolonged stress, and recurrent acute inflammation episodes, may have a significant role in the onset of diabetes [7,8], whereas human nutrition has a crucial role as a modifiable contributing factor. It appears that providing a diet high in antioxidants may help lower the risk of developing diabetes and enhance metabolic parameters in T2DM patients [9,10]. In the last few years, the development of polyphenol research concerning their impact on diabetes has markedly progressed. Numerous in vivo and in vitro studies provide knowledge for a better understanding of the role of nutrition and its impact on diabetes treatment [11,12,13,14]. Natural compounds, such as flavonoids, have been extensively researched for their ability to target glucose metabolism and inflammatory processes by influencing signaling cascades and immune system cell responses [15,16,17]. It is encouraging that some flavonoids have positive effects that are comparable to those of clinically used anti-diabetic medications [18].

Anthocyanins (ACNs) are a subclass of flavonoids and demonstrate potential for counteracting the onset and progression of diabetes. They have been shown to have positive effects by inhibiting digestive enzymes, enhancing insulin secretion, reducing apoptosis, promoting proliferation of pancreatic β-cells, and improving hyperglycemia through regulation of glucose metabolism in hepatocytes. Furthermore, they may decrease insulin resistance, inflammation, and oxidative stress in muscle and fat, while enhancing glucose uptake in both skeletal muscle and white adipose tissue [19,20]. It should be noted that ACNs possess antioxidant properties that can effectively inhibit the activation of pro-inflammatory pathways, which are often exacerbated by oxidative stress under diabetes conditions. Additionally, treatment with one of the most widely distributed ACNs, cyanidin-3-O-glucoside, accelerated the growth of blood vessels and collagen, showing that anthocyanins help diabetic wounds heal [21,22].

In the query performed in the ClinicalTrials.gov (accessed on 15 January 2024) database in February 2024, the term “anthocyanin” detected 127 registered clinical studies, of which 9 were related to diabetes. Therefore, this narrative review examines recent research on the therapeutic effects of dietary anthocyanins in treating diabetes, with a special focus on trials conducted in humans. The main objective of this current investigation was to revise and provide a more comprehensive assessment of the associations between the anti-diabetic effects of anthocyanins on humans, whether consumed as dietary supplements, purified substances, flavonoid mixes, or extracts.

## 2. Materials and Methods

For this study, the Medline database via PubMed research was searched with the combination of the following queries in titles and abstracts: (anthocyanins OR “anthocyanin-rich” OR pelargonidin OR cyanidin OR peonidin OR delphinidin OR petunidin OR malvidin OR cyanidin-3-O-glucoside OR berries OR cranberry OR aronia OR blueberry OR elderberry OR cherry) AND (diabetes OR ”glucose homeostasis” OR “insulin resistance” OR “glycemic control” OR fasting blood glucose” OR “FBG” OR “glycated hemoglobin” OR “HbA1c” OR “postprandial glucose” OR “homeostasis model assessment of insulin resistance” OR “insulin sensitivity” OR stroke OR blood pressure OR hypertension OR hyperlipidemia OR cholesterol OR triglycerides OR obesity OR “blood glucose” OR “endothelial function”). Using specific criteria, we restricted the article categories to randomized controlled trials (RCTs), clinical trials, and meta-analyses published in the English language within the last five years (from 1 January 2019 to 28 February 2024). The exclusion criteria refer to articles that present nonclinical outcomes, report insufficient methodology, and do not fulfill the inclusion criteria. The authors carefully assessed 18 studies (14 RCTs and 4 meta-analyses) using the bibliography management program EndNote X7. We imposed no publishing time limits on preclinical data. Nevertheless, we provided a revision of the latest studies (released in 2023) that the readers may not be familiar with.

## 3. Anthocyanins—Bioactivity and Metabolism

Anthocyanins are known as one of the water-soluble subclasses of flavonoids. These substances are responsible not only for the red to violet pigmentation of fruits and vegetables but also possess the ability to protect plants from the harmful effects of UV radiation [23,24]. ACNs are glucosides of the anthocyanidins. This form of anthocyanidin is more stable, less reactive, and has greater soluble capacity. Thus, this explains the exclusive occurrence of these substances in a glycosylated form. To date, over 700 anthocyanin derivatives have been identified, with only 27 having an aglycon structure (anthocyanidins) [25,26]. In a complex structure of anthocyanins, double benzoyl rings A and B are separated by a heterocyclic ring C. There are six different groups of these substances based on the number of hydroxyl and methoxyl groups attached to the B ring. These groups are cyanidin, delphinidin, pelargonidin, peonidin, malvidin, and petunidin. Cyanidins are predominant anthocyanidins in foods, accounting for 50% of all molecules [27,28].

The digestion process of anthocyanins starts in the oral cavity, where human saliva has the potential to degrade these substances from dietary sources. However, a larger proportion of dietary anthocyanins remain unabsorbed. According to the literature data, up to 65% of ACNs are not absorbed in the stomach and upper intestine, so they pass through the large intestine, and they interact with the gut bacteria [23,29]. Through gut microbial degradation, their bioavailability is potentially enhanced, and various metabolites are produced. After being absorbed into enterocytes, ACNs are metabolized by phase I and phase II enzymes. This produces phenolic acids and their conjugated products with additional hydroxyl, methyl, sulfuric, or glycoside groups [30,31].

Despite the relatively poor bioavailability of anthocyanins, the products of their catabolic breakdown exhibit high absorption [32]. This phenomenon emphasizes the potential importance of anthocyanin metabolites in providing health benefits. Results from animal and human studies showed that the bioavailability of anthocyanins depends on the physicochemical properties of each anthocyanin’s specific structure [29,33]. However, the complete explanation of intestinal absorption mechanisms, including their respective transport routes, remains unclear. The availability of novel methodologies, including in situ intestinal matrix-assisted laser desorption/ionization mass spectrometry imaging (MALDI-MS), contributes to understanding the bioavailability of anthocyanins. This experimental system enables the visualization of anthocyanin absorption and metabolism in real animal intestinal tissue without the use of specific markers, such as antibodies [34]. Hahm and colleagues investigated the absorption of both acylated and aglycon forms of anthocyanin from purple carrot *(Daucus carota* L.) extract [35]. The results demonstrated the absorption of acylated cyanidins in the rat jejunum membranes. The intestinal organic anion-transporting polypeptide 2B1 (OATP 2B1) pathway partially mediated these effects. Furthermore, both glucose transporter 2 (GLUT2) and OATP 2B1 were linked to the transportation of aglycon forms of cyanidins. This study provided the first evidence of how the intestines absorb acylated anthocyanins [34,35]. In addition, recent evidence demonstrates that acylated forms of anthocyanins may exert a stronger modulatory impact on inflammation, energy metabolism, and gut microorganisms in individuals with T2DM compared to their nonacylated forms [30,36].

## 4. Anthocyanins in the Human Diet

Anthocyanin components are prevalent in many fruits, particularly vegetables, nuts, and red wine (Table 1). According to published data, the content of ACNs in food may vary. It depends mostly on genetic, environmental, and agronomic endowments. Additionally, food processing and storage conditions influence the concentration of anthocyanins. Interestingly, variations in anthocyanin content between the same plant species are negligible. Both the United States Department of Agriculture (USDA) databases and the online Phenol-Explorer database (https://phenol-explorer.eu, accessed on 15 January 2024) contain data on anthocyanin contents in foods. However, in this dataset, there is still a paucity of data reporting the anthocyanin composition of particular products, i.e., black rice, Chinese cabbage, purple cauliflower, and black carrot [35,37,38,39]. Scientific publications provide specific information on the composition of these special products.

In Europe, the main sources of anthocyanins are mostly fruits like grapes, apples, pears, and berries (approximately 50%) and wines, whereas in the United States, berries, vines, grapes, and bananas represent around half of average daily anthocyanin intake [40,41]. Radishes, strawberries, persimmons, and grapes dominate Korean adults’ intake of ACNs [42]. Thus, the major dietary sources of anthocyanidins in the population of middle-aged Australian men and women are berries (24%), apples and pears (17%), and wine (12%) [43]. Different populations estimate the daily intake of anthocyanins in the human diet to range from a few milligrams to several hundred milligrams. Nevertheless, its estimation is still unlikely to represent all individuals and mostly depends on human food choices. Additionally, due to their advantageous impacts on human health, especially among subjects with above-normal values, anthocyanins may be consumed as supplements.

**Table 1 nutrients-16-01674-t001:** Presence of anthocyanins in specific foodstuffs (mg per 100 g of foodstuff) based on [44,45].

Fruits	ACNs (mg/100 g)	Vegetables	ACNs (mg/100 g)
Raspberries, black	685.70	Cowpeas, black seed cultivar, raw (*Vigna unguiculata* Subsp. *Sinensis*)	262.49
Plum, Illawara (*Podocarpus elatus*)	558.19	Cabbage, red, raw (*Brassica oleracea* (Capitata Group))	209.95
Chokeberry	349.79	Radicchio, raw (*Cichorium intybus*)	134.67
Bilberries	285.21	Eggplant, raw (*Solanum melongena*)	85.69
Service (Saskatoon) berries (*Amelanchier canadensis*)	180.78	Radishes (*Raphanussativus*)	63.13
Blueberries, cultivated (highbush) (*Vaccinium* spp.)	163.30	Black beans, mature seeds, raw (*Phaseolus vulgaris*)	44.52
Black currant (*Ribes nigrum*)	157.78	Wheat, purple	25.85
Blueberries, rabbiteye (*Vaccinium* spp.)	148.61	Nuts	ACNs (mg/100 g)
Grapes, Concord (*Vitis vinifera*)	120.10	Pecan nuts	18.02
Blackberries (*Rubus* spp.)	100.61	Pistachio nuts	7.33
Molucca raspberry (*Rubus moluccanus var. austropacificus*)	94.24	Hazel nuts	6.71
Maqui (Chilean wineberry) (*Aristotelia chilensis*)	88.52	Other products	ACNs (mg/100 g)
Red currants	75.02	Elderberry juice concentrate	411.40
Guajiru (coco-plum)	72.73	Sweet dessert wine	109.29
Acai berries, purple	53.64	Red table wine	19.27
Raspberries (*Rubus* spp.)	48.63		
Strawberries (*Fragaria X ananassa*)	27.01		

## 5. Recent Advances in Understanding Dietary Anthocyanins’ Anti-Diabetic Actions

Anthocyanins and their metabolites, which are present in foodstuff, exhibit a wide range of biochemical properties. Recent findings concerning anthocyanins’ anti-diabetic properties focus on studies investigating the cellular and molecular mechanisms involved. Numerous cell experiments and animal studies provide evidence for the favorable impact of anthocyanins on glucose regulation, gut microbiota improvement, and inflammation reduction under diabetic conditions (Table 2).

Current in vitro and animal studies have shown that anthocyanins are effective carbohydrate digestive enzyme inhibitors [13,46,47,48,49,50,51,52]. These mechanisms undoubtedly play a crucial role in their anti-diabetic properties, thereby reducing postprandial blood glucose levels. We must emphasize that well-known anti-diabetic medications, such as acarbose, exhibit similar anti-diabetic actions. Both anthocyanins and acarbose target α-glucosidase and pancreatic α-amylase breakdown, reducing the amount of glucose released into the bloodstream [11,46]. Surprisingly, this effect of anthocyanins from Aronia melanocarpa fruit extracts is many times higher than that elicited by acarbose [11,53,54].

Free fatty acid receptor 1 (FFAR1), a molecule activated by medium-to-long-chain fatty acids, is recognized as a stimulator of glucose-dependent insulin secretion in pancreatic β-cells. At the cellular level, activated FFAR1 increases calcium ions mobility and insulin and glucagon production [55]. In addition, activation of FFAR1 stimulates the production of the peptide YY (PYY), thereby leading to the control of body weight [56]. Studies have shown that dietary anthocyanins from purple corn activate FFAR1, thereby enhancing insulin secretion and hepatic glucose uptake [57]. Treatment with 1 mg/mL of purple corn anthocyanin-rich water extract (PCW) increased glucose-stimulated insulin secretion in INS-1E cells by 52% and increased glucose uptake in HepG2 cells by 48%. Furthermore, PCW was also involved in the activation of glucokinase (GK), which is a well-known glucose sensor in the β-cells of the pancreas. This study suggested that ANCs from purple corn may exert anti-diabetic effects in vivo.

It is increasingly recognized that low-grade chronic inflammation (LGCI) is a pathological core feature of diabetes [58,59]. Obesity-related increased fat tissue plays a crucial role in increased plasma levels of proinflammatory cytokines and, therefore, the occurrence of T2DM and its complications. That phenomenon has been termed “Diabesity”, which refers to the adverse health consequences of both T2DM and obesity or overweight. Recently, it was reported that fourteen-day maqui berry treatment reduced weight gain, blood fasting glucose, and insulin resistance in obese diabetic rats, which were used as a model of metabolic syndrome [60]. The administration of maqui berries reduced the concentration of plasma malondialdehyde (MDA), a well-known lipid peroxidation product, while simultaneously increasing the activity of the antioxidant enzyme superoxide dismutase (SOD). Similarly, the beneficial effects of anthocyanins were reported in cell-culture studies [49,50]. The anthocyanin-rich extract from Pitaya (*Hylocereus lemairei*) was effective in attenuating oxidative stress in human enterocytes under high glucose concentrations [51]. In a similar study, Zhu et al. evaluated the metabolic effects of five blueberry anthocyanins in high glucose and oleic acid-treated HepG2 cells [49]. Five anthocyanins isolated from rabbiteye blueberry (*Vaccinium virgatum*) demonstrated antioxidant effects by exhibiting oxygen radical absorbance capacity (ORAC) and the scavenging capacity of 2,2-diphenyl-1-picrylhydrazyl (DPPH) and azinobis-3-ethylbenzothiazoline-6-sulfonic acid (ABTS) free radicals. Additionally, four of them enhanced glucose uptake and reduced lipid accumulation [49]. These findings further show that ANCs regulate glucose homeostasis by modifying inflammation and lipid metabolism.

Evidence from numerous investigations has shown that elevated levels of reactive oxygen species (ROS) in the bloodstream resulting from deposited fat have been linked to activating insulin resistance (IR) in various adipose tissues and skeletal muscles [58,61]. IR is a common feature of type 2 diabetes, and resistance occurs long before the onset of the disease. Anthocyanins were demonstrated to inhibit insulin resistance. They may exert these effects by enhancing hepatic protein expression levels of the phosphorylated phosphoinositide 3-kinases (PI3Ks), protein kinase (Akt), and glycogen synthase kinase-3 (GSK3β) proteins essential for maintaining key hypoglycemic signaling pathways. Anthocyanins were also associated with suppressed expression of protein tyrosine phosphatase 1B (PTP1B), which is involved in glucose metabolism, and its depletion or inhibition has potential benefits in managing diabetes [62,63,64]. Interestingly, in insulin-resistant HepG2 cells and a diabetic mouse model, a combination of metformin and cyanidin-3-O-arabinoside (C3A) treatment enhanced the inhibitory effect of C3A on PTP1B, thereby confirming its synergistic effects [62]. These effects may be explained as complementary mechanisms of action. Primarily metformin reduces the synthesis of hepatic glucose production and enhances insulin sensitivity, while ACNs further promote insulin signaling pathways and provide additional antioxidant protection. 

In recent years, a growing body of research has shown that intestinal microbiota dysbiosis is closely associated with the development of type 2 diabetes mellitus [7,19,30,65,66]. An imbalance in the diversity of microbiota might harm the integrity of the gut barrier. This action induces chronic low-grade inflammation linked to insulin resistance, adiposity, and de novo synthesis of triglycerides. A change in the proportion of Firmicutes/Bacteroidetes is observed in diabetes model mice. Diabetes mice appeared to have fewer phylum Bacteroidetes and more Firmicutes compared with non-diabetic mice [30,62]. Huang et al. verified that a supplementation diet with 7.2 mg/kg/day of cyanidin-3-glucoside (C3G) was able to increase the Bacteroidetes/Firmicutes phylum bacteria ratio and the abundance of gut Muribaculaceae family bacteria in a diet-induced insulin-resistant mouse model [67]. Furthermore, C3G improved the reduction in gut microbial genes involved in inflammation and enhanced gut microbial genes involved in metabolic processes. In other works, the black rice and black bean husk anthocyanin-rich extracts modified the T2DM rat intestinal microbiota by enhancing the abundance of short-chain fatty acid (SCFA), thereby inducing the growth of common beneficial *Akkermansia* spp., *Phascolarctobacterium* spp., *Bacteroides* spp., and *Coprococcus* spp. [68]. Also, in a recent study, combined metformin and anthocyanin treatment had a positive regulatory effect on the intestine by increasing the abundance of the Lactobacillus and Bifidobacterium phylum [62]. These studies demonstrate that anthocyanins may exert their anti-diabetic effects by modulating microbial populations, thereby improving their richness and the proportion of gut microbes’ beneficial populations.

**Table 2 nutrients-16-01674-t002:** In vivo and in vitro research update (studies published in 2023) on anthocyanins and their anti-diabetic actions.

Actions/Substances	In Vitro, In Vivo, or In Silico Model	Mode of Action	References
delphinidin-3-O-galactoside, delphinidin-3-O-glucoside, petunidin-3-Ogalactoside, petunidin-3-O-glucoside, and malvidin-3-O-galactoside) isolated from rabbiteye blueberry (*Vaccinium virgatum*)	HepG2 cells	↑ glucose uptake inhibiting activity of α-glucosidase exhibiting ORAC scavenging power of ABTS+, and DPPH-free radical	[49]
*Prunus lusitanica*, cyanidin 3-glucoside	HepG2, RAW 264.7, Caco-2 cells	inhibiting NO release inhibiting α-glucosidase	[50]
Cyanidin 3-(p-coumaroyl)-diglucoside-5-glucoside, Malvin, Nasunin, cyanidin 3-O-xylosyl-rutinoside, and cyanidin 3-O-rutinoside	molecular docking, integrated computer-aided approach	inhibiting PTP1B, DPP4, α-amylase	[64]
Maqui berry (*Aristotelia chilensis*), delphinidin	MetS male and female rats	↓ weight gain ↓ blood fasting glucose ↓ TC, TGs ↓ IR ↑ BP, SOD activity ↓ MDA	[60]
Cyanidin-3-O-glucoside	isolated mouse islets and the INS-1E cell	↓ CHOP expression	
Bilberry (*Vaccinium myrtillus*), elphinidin-3-galactoside, and malvidin-3-glucoside	alpha-amylase enzyme	inhibiting of α-amylase	[13]
Pitaya (*Hylocereus lemairei*)	human enterocytes under high glucose concentration	↓ oxidative stress ↓ NO inhibiting α-glucosidase and pancreatic lipase strong redox capacity	[51]
Hibiscus rosa-sinensis flower anthocyanin-rich extract	in vitro	inhibiting maltase, sucrase, isomaltase, glucoamylase, and AChE	[52]
Anthocyanin and metformin	insulin-resistant HepG2 cells and a diabetic mouse model	synergistic restorative effects on the blood glucose level, IR, and organ damage in the liver, pancreas, and ileum ↑ short-chain fatty acid ↑ beneficial bacteria suppressing protein tyrosine phosphatase 1B expression regulating the PI3K/Akt/GSK3β pathway	[62]

↑—increase; ↓—decrease; ABTS—2,2′-azinobis-3ethylbenzothiazoline-6-sulfonic acid; AChE—acetylcholinesterase; Akt—protein kinase; BP—blood pressure; CHOP—C/EBP homologous protein; DPP4—dipeptidyl-peptidase-4; DPPH—2,2-diphenyl-1-picrylhydrazyl; GSK3β—glycogen synthase kinase-3; IR—insulin resistance; MDA—malondialdehyde; MetS—metabolic syndrome; NO—nitric oxide; ORAC—oxygen radical absorbance capacity; PI3K—phosphorylated phosphoinositide 3-kinase; PTP1B—protein tyrosine phosphatase 1B; SOD—superoxide dismutase; TC—total cholesterol; TG—triacylglycerides.

## 6. Clinical Studies on Anthocyanin Interventions—An Update from the Last 5 Years

In recent years, there has been increasing attention on the use of natural bioactive compounds in the prevention and management of metabolic disorders, such as T2DM. In addition to medication, human diet is essential for controlling blood glucose levels in diabetic patients. A meta-analysis of eight prospective cohort studies, involving 394,913 participants, revealed promising data linking a dietary anthocyanin intake to a 15% lower incidence of type 2 diabetes mellitus. Furthermore, there was a 5% decrease in the occurrence of T2DM when the dietary intake of anthocyanins increased by 7.5 mg/day or when the intake of berries increased by 17 g/day [69]. Furthermore, the anti-diabetic actions of ACNs have been widely investigated in pre-clinical studies; however, the data collected in humans remain limited. The current randomized controlled human studies (Appendix A) have focused on the anti-diabetic benefits of anthocyanins and their influence on glucose metabolism in both individuals at risk of developing type 2 diabetes mellitus and those already diagnosed with T2DM. In this update, we focus on clinical research conducted within the last five years that provides the most compelling evidence for the preventive role of anthocyanin consumption in the development of T2DM.

### 6.1. Anthocyanins and Glycemic Status 

The effect of either pure anthocyanins or fruit extracts rich in anthocyanins on the fasting blood glucose (FBG) of T2DM individuals was widely investigated [47,70,71,72]. Tasic et al. demonstrated a significant reduction in fasting blood glucose (FBG) after 2- and 4-week standardized Aronia L. Melanocarpa extract treatment in patients with MetS and confirmed T2DM [73]. In a similar study, daily oral administration of 320 mg/day anthocyanins (MEDOX^®^ capsules) for four weeks reduced FBG only in the T2DM at-risk group but not in T2DM and healthy individuals [71]. Twelve-week dietary supplementation with 320 mg of pure anthocyanins per day significantly reduced HbA1c levels in 76 patients with prediabetes or newly diagnosed diabetes but did not significantly improve fasting blood glucose levels [70]. A similar study showed significant reductions in HbA1c in T2DM patients who supplemented their diet with 150 mL of chokeberry juice for three months [74]. 

Not all results from clinical trials provide evidence of the protective effects of anthocyanin on glucose levels in diabetic individuals. Consuming fermented or non-fermented aronia extract bars (893 mg or 533 mg of anthocyanins) for 8 weeks did not improve plasma FBG in patients with T2DM [75]. The same study found no improvement in the levels of glycated hemoglobin (HbA1C), a glycemic control indicator. The relatively well-controlled group of patients’ diabetes risk markers could explain the discrepancy between these reports. In contrast to these results, data from all meta-analyses of randomized control trials from the last 5 years reported a reduction in both fasting blood glucose and glycated hemoglobin levels after the supplementation of pure or natural sources of anthocyanins [76,77,78,79,80]. In a meta-analysis of 22 randomized controlled trials, blueberry and cranberry intake reduced FBG by 17.72 mg/dL and reduced glycated hemoglobin by 0.32% in humans with T2DM [78]. According to the meta-analyses of Fallah et al., supplementation with higher doses of anthocyanins (>300 mg/day) for more than 8 weeks significantly had more favorable effects on FBG and HbA1c levels. Moreover, ACN intake from fruit extracts or powders showed a significantly higher effect on lowering HbA1c than that of pure anthocyanin [76]. Altogether, these data suggest a potentially significant beneficial effect of daily anthocyanin intake.

### 6.2. Insulin Resistance and Inflammation

In line with the in vitro and in vivo studies, the significant effects of anthocyanins on insulin resistance and inflammation were also investigated in human clinical studies. Dysregulation of carbohydrate metabolism, a leading pathogenic factor for T2DM, always accompanies metabolic inflammation and the weakening of cells’ sensitivity to insulin. Targeting inflammatory pathways could be a component of strategies to prevent and control diabetes and related complications. Favorable effects of the anthocyanin combination (Medox^®^) were reported in a study conducted among 40 patients with type 2 diabetes, T2DM at-risk individuals, and healthy individuals. Over 4 weeks, supplementation had beneficial effects on biomarkers of inflammation in the T2DM group. In diabetic patients, daily supplementation with 320 mg of ACNs was found to inhibit the expression of proinflammatory factors related to NF-κB pathways, including TNF-α, IL-6, and IL-18 [71]. At the cellular level, the activated transcription factor NF-κB induced the activity of TNF-α and the interleukins, thereby altering insulin sensitivity [46,61]. Therefore, the use of anthocyanin, which has been shown to have positive effects on the process of inflammation, may contribute to the development of treatment approaches focused on inhibiting NF-κB.

Solverson et al. analyzed the influence of consuming a combination of whole mixed berries or mixed berry juice on insulin sensitivity [81]. The study included a fully controlled, high-fat diet consisting of one of four treatment foods: whole mixed berries, pressed mixed berry juice, sugar-matched gelatin, or sugar-matched fiber-enriched gelatin. No significant differences were observed comparing all of the treatment options. Nevertheless, a second analysis revealed that berry treatments showed an increase in blood insulin levels and a notable trend toward a reduction in serum glucose. This suggests that mixed berries may have glucoregulatory effects and lower the risk of diabetes in overweight or obese adults consuming a high-fat diet [81]. In a meta-analysis of 37 randomized controlled studies, Fallah evaluated the effects of anthocyanins on the homeostasis model assessment of insulin resistance (HOMA-IR) and biomarkers of glycemic control [77]. The results showed that consuming pure anthocyanins or anthocyanin-rich meals did not have a significant impact on serum insulin levels while significantly reducing the HOMA-IR index in individuals with type 2 diabetes and those who were overweight or obese.

Not all studies provide significant effects of ACNs on HOMA-IR levels in diabetes patients and subjects at high diabetic risk [76,78,79,82,83]. Apart from a significant reduction in FBG at a median dose of 320 mg/day, anthocyanin intake had no beneficial effect on HOMA-IR, as well as fasting insulin levels, according to a meta-analysis by Mao, which included 703 individuals with T2DM from thirteen RCTs [76]. Similarly, a recent meta-analysis of 27 randomized controlled studies reported that polyphenol-rich mixtures, including ACNs administrated for 1 week to one year, had no significant effect on HOMA-IR and fasting insulin levels. Nevertheless, the main limitation of this study is the existence of diverse bioactive substances in the extracts, which may have varying effects on the insulin profile [79]. A current clinical investigation showed that regular aronia bar consumption, rich in anthocyanins, for 8 weeks did not lead to improvement in fasting insulin levels in T2DM individuals. However, there was a significant increase in glucose-dependent insulinotropic peptide (GIP) levels, suggesting that ACNs may promote insulin secretion from pancreatic beta cells by modulating this regulatory hormone of insulin secretion. Given these data, it is tempting to speculate that anthocyanins could exert anti-diabetic benefits through glycemic control. However, further randomized controlled trials are necessary to provide a more comprehensive knowledge of the role of ACNs in insulin resistance in individuals with T2DM.

### 6.3. Anthocyanins and Lipidemic Status in T2DM

Several lines of evidence have shown that long-term hyperglycemia leads to the development of dyslipidemia in patients diagnosed with T2DM through a process including the production of glycosylation end products from non-enzymatic interactions between glucose and proteins or lipoproteins. Thus, targeting dyslipidemia may improve diabetes treatment [69]. 

Recently, ACN-rich products were linked to reducing dyslipidemia in T2DM patients [72,73,76,80,82,83,84]. Four-week dietary supplementation with Alixir 400 PROTECT^®^ (Pharmanova, Belgrade, Serbia) (120 mg of anthocyanins/day) improved total cholesterol (TC), LDL-cholesterol (LDL-c), triglycerides (TGs), and HDL-cholesterol (HDL-c) in 143 individuals with both MetS and MetS and confirmed diabetes [73]. Furthermore, the regular intake of 30 mL Montmorency tart cherry juice reduced TC and LDL-c levels by 9.01% and 11.72%, respectively, in patients at risk of diabetes [72]. Interestingly, an increased level of HDL-c was observed in a group of patients with diabetes supplemented with anthocyanin-rich bars. However, in the same study, an unexpected increase in triglyceride levels was observed [82]. Finally, data from two meta-analyses of RCTs involving patients with T2DM reported that chronic supplementation with pure anthocyanins or anthocyanin-rich extracts had a beneficial effect on blood lipid levels [76,80]. Collectively, these studies demonstrate the hypolipidemic abilities of anthocyanins in T2DM and T2DM at-risk patients.

### 6.4. Effects of Anthocyanins on Other Parameters Related to Diabetes

Recent clinical studies have also evaluated anthocyanin’s effects on endothelial function (EF), uric acid levels, and microbial metabolites in both prediabetes and diabetes patients [71,83,85,86].

Endothelial dysfunction is the first manifestation in the development of hypertension and cardiovascular disease. In a recent clinical study, intake of high-polyphenolic extra virgin olive oil together with ½ cup of frozen blueberries enhanced EF in adults at risk of T2DM [85]. Similarly, the daily administration of 150g of blueberries per six months to 115 adults with MetS resulted in significant improvements in endothelial function, as well as systemic arterial stiffness [83].

Uric acid is considered a significant risk factors for T2DM, as the concentration of serum urate is correlated with blood glucose levels [87]. Anthocyanins are suggested as hypouricemic agents. Nikbakht et al. reported that uric acid levels significantly decreased from 334.5 ± 22.71 umol/L to 281.0 ± 14.33 umol/L in the T2D at-risk group, which supplemented 320 mg of anthocyanins per four weeks [71]. In another study, regular intake of red raspberry (*Rubus idaeus* L.) with fructo-oligosaccharide increased the microbial metabolites of polyphenols in the blood after 4 weeks of supplementation in both healthy adults and adults with prediabetes and insulin resistance [86]. Altogether, these results further show the favorable effects of anthocyanins on other parameters related to diabetes.

## 7. Summary

This narrative review presents a comprehensive summary of findings from randomized controlled trials investigating the anti-diabetic properties of dietary anthocyanins. During the last five years, a minimum of 18 clinical investigations have examined the correlation between the intake of ACNs and the clinical and biochemical parameters related to diabetes. The results obtained from the clinical studies demonstrate that anthocyanin could simultaneously affect many targets associated with T2DM. Furthermore, ACN intake during diabetes or hyperglycemia appears to be safe and effective. Nevertheless, the limitations of the presented studies were also observed. The primary constraints of the clinical data provided are the limited duration of investigation, small sample sizes, and the various types and doses of anthocyanin supplements used. Moreover, the various effects of anthocyanins and other bioactive components, used together in many interventions, may restrict the current findings. Furthermore, the majority of randomized controlled trials did not adjust the intake of anthocyanins from dietary sources, potentially impacting the outcomes.

Regarding anthocyanins’ effects on glycemic control, future research should focus on the optimal intervention duration. We should emphasize that alleviating the clinical symptoms of diabetes is a gradual process. For example, the glycation of hemoglobin is used to measure the mean blood glucose levels throughout a prior period of 2 to 3 months. Future investigations should also take into account the use of multiple clinical biomarkers to monitor diabetes and pancreatic function. Thus, the examination of the potential protective action of anthocyanins on diabetes outcomes would be more complex. Studies that consider variability in modern drug effectiveness in diabetes treatment are scarce. In future studies, there is a need to analyze anthocyanin effects among well-controlled versus poorly controlled groups of patients with diabetes, especially since the beneficial effects of anthocyanin use were most prominent in the group of patients with above-normal biomarkers of glycemic control. Finally, further investigation should consider interindividual differences in anthocyanin metabolism and the interactions between nutrients and the microbial flora.

In conclusion, this review provides evidence that an anthocyanin-rich diet can improve diabetes outcomes, especially in a group at risk. As a result, supplementing with additional amounts of ACNs has the potential to delay the progression of T2DM. The current data would help medical professionals understand the beneficial role of anthocyanins in the prevention and treatment of type 2 diabetes mellitus.

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
