# Peer review of "Anthocyanins and Type 2 Diabetes: An Update of Human Study and Clinical Trial"

_nutrients, 2024, doi:10.3390/nu16111674_

Round 1

Reviewer 1 Report

Comments and Suggestions for Authors

Title: Anthocyanins and Type 2 Diabetes. An Update of Human 2 Study and Clinical Trial

 The manuscript discusses the potential therapeutic effects of anthocyanins on type 2 diabetes based on pre-clinical and clinical studies. It suggests that anthocyanins could play a significant role in regulating glucose levels, improving gut microbiota, and reducing inflammation in diabetic conditions. The review also highlights the need for further research to better understand the optimal use of anthocyanins in diabetes management.

 Here are some general recommendations to improve your writing:

The introduction provides a valuable overview of the effects of anthocyanins on diabetes. However, improvements in the introduction and discussion sections could strengthen the overall impact of the review. It is necessary to go deeper into the pathophysiological basis of type 2 diabetes to explain more rationally the effect of anthocyanins.

Table 1 provides basic information and could be enhanced by including the effects of anthocyanins on diabetes.

In general, the text contains errors in the use of italics for terms such as "in vivo" or "in vitro".

Please ensure consistency in the use of abbreviations, such as "Type 2 diabetes mellitus," throughout the document. (e.g line 239).
It is recommended to create figures that provide a concise summary of the findings presented in this document.

Author Response

Response to Reviewer 1.

Reviewer 2 Report

Comments and Suggestions for Authors

Comments to the Authors of manuscript number nutrients-3012946 entitled “Anthocyanins and Type 2 Diabetes. An Update of Human Study and Clinical Trial.

Anthocyanins, found in fruits and vegetables, have been shown in seven pre-clinical studies to regulate glucose levels, improve gut microbiota, and reduce inflammation in diabetic conditions. This review summarizes recent clinical research on the therapeutic effects of dietary anthocyanins for diabetes management, highlighting their positive impact on fasting blood glucose, glycated hemoglobin, and other diabetes indicators. Higher anthocyanin dosages were particularly beneficial. It is a very interesting review.

1. Introduction states that anthocyanins have shown beneficial effects in various aspects of diabetes management, it is important to note that these findings are based on a limited number of studies. The efficacy of anthocyanins might vary among different populations and study designs.

It is suggested that anthocyanins can have effects comparable to clinically used anti-diabetic drugs.

It describes several mechanisms by which anthocyanins may exert their effects (e.g., inhibiting digestive enzymes, enhancing insulin secretion, reducing apoptosis). However, the extent to which these mechanisms contribute to the overall therapeutic effects in humans is still not fully understood.

Anthocyanin supplementation is mentioned in various forms (dietary supplements, pure compounds, flavonoid mixtures, or extracts), but it does not specify the optimal dosage and form that would be most effective for diabetes management.

2. Material and methods are described properly.

3. the catabolic breakdown products of anthocyanins exhibit high absorption, the specific activities and health benefits of these metabolites compared to the parent compounds need be mentioned

4. The text mentions that maqui berry treatment reduced various metabolic parameters in obese diabetic rats. However, the translation of these results to human health requires rigorous clinical trials to confirm efficacy and safety in humans.

5. what about synergistic effects with metformin?

6. There is mention of an unexpected increase in triglyceride levels in a group of patients consuming anthocyanin-rich bars, despite improvements in other lipid parameters. This inconsistency suggests that the effects of anthocyanins on lipid metabolism may not be fully understood.

7. an anthocyanin-rich diet can improve diabetes outcomes, especially in at-risk groups, and supplementing with anthocyanins may delay T2DM progression. This information is valuable for medical professionals in understanding the role of anthocyanins in diabetes prevention and management.

8. Include in the methodology which authors searched for articles.

9. Some typos are present in the text. For example:

Line 80 “familiar with” add “.”

Line 349 “anthocyaninsin”

Line 395 “poorly controlled”

Author Response

Respond to Reviewer 2.

Reviewer 3 Report

Comments and Suggestions for Authors

Thank you very much for allowing me to review the review article titled “Anthocyanins and Type 2 Diabetes. An Update of Human Study and Clinical Trial “ (nutrients-3012946). The aim of this review is to update and provide a more comprehensive estimate of the association between anthocyanin intake, either as dietary supplements or pure compounds, flavonoid mixtures, or extracts, and their anti-diabetic effects in humans.

Comments:

The title of the work aligns with the content of the manuscript. However, since it is a review, this should be indicated in the title.

Regarding the abstract, which is very well written, I believe it is essential to include the type of review conducted. According to the objectives, it is a comprehensive review. The period covered by the review should also be mentioned to connect with other past and future reviews on this topic. Additionally, the minimum number of articles included in the review should be specified, as this information is crucial for understanding the scope of the review.

In relation to the introduction, the section from lines 46-55, which describes the characteristics of anthocyanins, should also highlight their antioxidant role.

In section 56-62, I believe lines 57, 58, and 59 should be moved to the Materials and Methods section instead of the introduction.

Prior to stating the objective, the underlying hypothesis of the review can be presented.

In the Materials and Methods section, line 64 should start by identifying the study design, indicating that it is a comprehensive review. I suggest the authors use a figure for article selection, such as the PRISMA diagram, or another appropriate model to better illustrate the article selection process.

Regarding line 80, it seems the sentence is incomplete. Please review this.

Starting from section 3, the results are presented. I suggest that section 3 be titled "Results" and the various evaluated aspects be presented as subsections.

The content of the review is indeed very interesting and updates the existing literature by integrating relevant information.

The summary presents both current knowledge and its limitations, highlighting the need for further research on this topic.

Author Response

Respond to Reviewer 3.
